# Object-Oriented Model Learning through Multi-Level Abstraction

## Abstract

Object-based approaches for learning action-conditioned dynamics has demonstrated promise for generalization and interpretability. However, existing approaches suffer from structural limitations and optimization difficulties for common environments with multiple dynamic objects. In this paper, we present a novel self-supervised learning framework, called *Multi-level Abstraction Object-oriented Predictor* (MAOP), for learning object-based dynamics models from raw visual observations. MAOP employs a three-level learning architecture that enables efficient dynamics learning for complex environments with a dynamic background. We also design a spatial-temporal relational reasoning mechanism to support instance-level dynamics learning and handle partial observability. Empirical results show that MAOP significantly outperforms previous methods in terms of sample efficiency and generalization over novel environments that have multiple controllable and uncontrollable dynamic objects and different static object layouts. In addition, MAOP learns semantically and visually interpretable disentangled representations.

## 1 Introduction

Model-based deep reinforcement learning (DRL) has recently attracted much attention for improving sample efficiency of DRL, such as (Heess et al., 2015; Schmidhuber, 2015; Gu et al., 2016; Racanière et al., 2017; Finn & Levine, 2017). One of the core problems for model-based reinforcement learning is to learn action-conditioned dynamics models through interacting with environments. Pixel-based approaches have been proposed for such dynamics learning from raw visual perception, achieving remarkable performance in training environments (Oh et al., 2015; Watter et al., 2015; Chiappa et al., 2017).

To unlock sample efficiency of model-based DRL, learning action-conditioned dynamics models that generalize over unseen environments is critical yet challenging. Finn et al. (2016) proposed an action-conditioned video prediction method that explicitly models pixel motion and thus is partially invariant to object appearances. Zhu et al. (2018) developed an object-oriented dynamics predictor, taking a further step towards generalization over unseen environments with different object layouts. However, due to structural limitations and optimization difficulties, these methods do not learn and generalize well for common environments with a dynamic background, which contain multiple moving objects in addition to controllable objects.

To address these limitations, this paper presents a novel self-supervised, object-oriented dynamics learning framework, called *Multi-level Abstraction Object-oriented Predictor* (MAOP). This framework simultaneously learns disentangled object representations and predicts object motions conditioned on their historical states, their interactions to other objects, and an agent's actions. To reduce the complexity of such concurrent learning and improve sample efficiency, MAOP employs a three-level learning architecture from the most abstract level of motion detection, to dynamic instance segmentation, and to dynamics learning and prediction. A more abstract learning level solves an easier problem and has lower learning complexity, and its output provides a coarse-grained guidance for the less abstract learning level, improving its speed and quality of learning convergence. This multi-level architecture is inspired by humans' multi-level motion perception from cognitive science studies (Johansson, 1975; Lu & Sperling, 1995; Smith et al., 1998) and multi-level abstraction search in constraint optimization (Zhang & Shah, 2016). In addition, we design a novel CNN-based

spatial-temporal relational reasoning mechanism, which includes a Relation Net to reason about spatial relations between objects and an Inertia Net to learn temporal effects. This mechanism offers a disentangled way to handle physical reasoning in the setting with partial observability.

Empirical results show that MAOP significantly outperforms previous methods in terms of sample efficiency and generalization over novel environments that have multiple controllable and uncontrollable dynamic objects and different object layouts. It can learn from few examples and accurately predict the dynamics of objects as well as raw visual observations in previously unseen environments. In addition, MAOP learns disentangled representations and gains semantically and visually interpretable knowledge, including meaningful object masks, accurate object motions, disentangled reasoning process, and the discovery of the controllable agent.

## 2 RELATED WORK

**Object-oriented reinforcement learning** has received much research attention, which exploits efficient representations based on objects and their interactions. This learning paradigm is close to that of human cognition in the physical world and the learned object-level knowledge can be robustly generalized across environments. Early work on object-oriented RL requires explicit encodings of object representations, such as *relational MDPs* (Guestrin et al., 2003), *OO-MDPs* (Diuk et al., 2008), object focused q-learning (Cobo et al., 2013), and Schema Networks (Kansky et al., 2017). In this paper, we present an end-to-end, self-supervised neural network framework that automatically learns object representations and dynamics conditioned on actions and object relations from raw visual observations.

**Action-conditioned dynamics learning** aims to address one of the core problems for model-based DRL, that is, constructing an environment dynamics model. Several pixel-based approaches have been proposed for learning how an environment changes in response to actions through unsupervised video prediction and achieve remarkable performance in training environments (Oh et al., 2015; Watter et al., 2015; Chiappa et al., 2017). Fragkiadaki et al. (2016) propose an object-centric prediction method to learn the dynamics model when given the object localization and tracking. Finn et al. (2016) develop a dynamics prediction method that explicitly models pixel motions and is partially invariant to object appearances, and its usage for model-based DRL is demonstrated with model predictive controller (Finn & Levine, 2017). Recently, Zhu et al. (2018) propose an object-oriented dynamics learning paradigm that enables its learned model to generalize over unseen environments with different object layouts and be robust to changes of object appearances. However, this paradigm focuses environments with a single dynamic object. In this paper, we take a further step towards learning object-oriented dynamics model in more general environments with multiple controlled and uncontrollable dynamic objects. In addition, we design an instance-aware dynamics mechanism to support instance-level dynamics learning and handle partial observations.

**Relation-based deep learning approaches** make significant progress in a wide range of domains such as physical reasoning (Chang et al., 2016; Battaglia et al., 2016), computer vision (Watters et al., 2017; Wu et al., 2017), natural language processing (Santoro et al., 2017), and reinforcement learning (Zambaldi et al., 2018; Zhu et al., 2018). Relation-based nets introduce relational inductive biases into neural networks, which facilitate generalization over entities and relations and enables relational reasoning (Battaglia et al., 2018). OODP (Zhu et al., 2018) is similar to this paper, learning object-level dynamics conditioned on actions and object-to-object relations. In contrast to OODP, this paper proposes a novel spatial-temporal relational reasoning mechanism, which includes an Inertia Net for learning temporal effects in addition to a CNN-based Relation Net for reasoning about spatial relations. This mechanism offers a disentangled way to handle physical reasoning in the setting of partial observability.

**Instance Segmentation** has been one of the fundamental problems in computer vision. Instance segmentation can be regarded as the combination of semantic segmentation and object localization. Many approaches have been proposed for instance segmentation, including DeepMask (Pinheiro et al., 2015), InstanceFCN (Dai et al., 2016), FCIS (Li et al., 2017), and Mask R-CNN (He et al., 2017). Most models are supervised learning and require a large labeled training dataset. Liu et al. (2015) proposes a weakly-supervised approach to infer object instances in foreground by exploiting dynamic consistency in video. In this paper, we design a self-supervised, three-level approach for learning dynamic rigid object instances. At the most abstract level, the foreground detection

module provides a coarse-grained guidance for producing region proposals at the instance segmentation level. The instance segmentation level then learns coarse instance segmentation. This coarse instance segmentation provides a guidance for learning accurate instances at the dynamics learning level, whose instance segmentation considers not only object appearances but also motion prediction conditioned on object-to-object relations and actions.

# 3 MULTI-LEVEL ABSTRACTION OBJECT-ORIENTED PREDICTOR (MAOP)

In this section, we will present a novel self-supervised deep learning framework, aiming to learn object-oriented dynamics models that are able to generalize over unseen environments with different object layouts and multiple dynamic objects. Such an object-oriented dynamics learning approach requires simultaneously learning object representations and motions conditioned on their historical states, their interactions to other objects, and an agent's actions. This concurrent learning is very challenging for an end-to-end approach in complex environments with a dynamic background. Evidences from cognitive science studies (Johansson, 1975; Lu & Sperling, 1995; Smith et al., 1998) show that human beings are born with prior motion perception ability (in Cortical area MT) of perceiving moving and motionlessness, which enables learning more complex knowledge, such as object-level dynamics prediction. Inspired by these studies, we design a multi-level learning framework, called *Multi-level Abstraction Object-oriented Predictor* (MAOP), which incorporates motion perception levels to assist in dynamics learning.

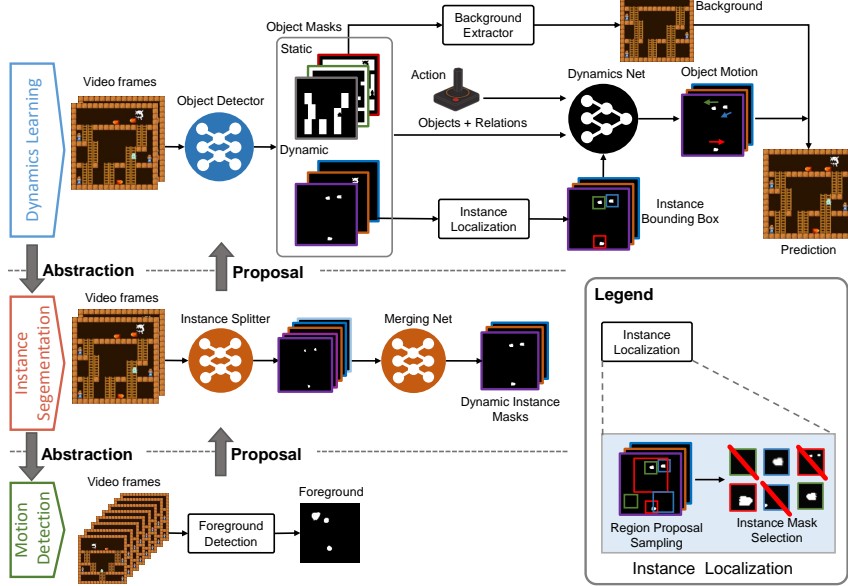

Figure 1: Multi-level abstraction framework from a top-down decomposition view. First, we perform motion detection to produce foreground masks. Then, we utilize the foreground masks as dynamic region proposals to guide the learning of dynamic instance segmentation. Finally, we use the learned dynamic instance segmentation networks (including Instance Splitter and Merging Net) as a guiding network to generate region proposals of dynamic instances and guide the learning of Object Detector in the level of dynamics learning.

Figure 1 illustrates three levels of MAOP framework: dynamics learning, dynamic instance segmentation, and motion detection. The dynamics learning level is an end-to-end, self-supervised neural network, aiming to learn object representations and instance-level dynamics, and predict the next visual observation conditioned on an agent's action. To guide the learning of the object representations and instance localization at the level of dynamics learning, the more abstracted level of dynamic instance segmentation learns a guiding network in a self-supervised manner, which can provide coarse dynamic instance mask proposals. It exploits the spatial-temporal information of locomotion property and appearance pattern to capture the region proposals of dynamic instances. To facilitate the learning of dynamic instance segmentation, MAOP employs the more coarse-grained level of motion

detection, which detects changes in image sequences and provides guidance on proposing regions potentially containing dynamic instance. As the learning proceeds, the knowledge distilled from the more coarse-grained level are gradually refined at the more fine-grained level by considering additional information. When the training is finished, the coarse-grained levels of dynamic instance segmentation and motion detection will be removed at the testing stage. In the rest of this section, we will describe in detail the design of each level and their connections.

## 3.1 OBJECT-ORIENTED DYNAMICS LEARNING LEVEL

The semantics of this level is formulated as learning an object-based dynamics model with the region proposals generated from the more abstracted level of dynamic instance segmentation. The whole architecture is shown in the top part of Figure 1, which is an end-to-end neural network and can be trained in a self-supervised manner. It takes a sequence of video frames and an agent's actions as input, learns the disentangled representations (including objects, relations and effects) and the dynamics of controllable and uncontrollable dynamic object instances conditioned on the actions and object relations, and produce the predictions of raw visual observations. The whole architecture includes four major components: A) an Object Detector that decomposes the input image into objects; B) an Instance Localization module responsible for localizing dynamic instances; C) a Dynamics Net for learning the motion of each dynamic instance conditioned on the effects from actions and object-level spatial-temporal relations; and D) a Background Constructor that computes the background image from the learned static object masks. Algorithm 1 illustrates the interactions of these components and the learning paradigm of object based dynamics, which is a general framework and agnostic to the concrete form of each component. In the following paragraphs, we will describe the detailed implementation of every components.

---

**Algorithm 1** Basic paradigm of object-oriented dynamics learning.

---

**Input:** A sequence of video frames $\mathbf{I}^{t-h:t}$ with length $h$, input action $\boldsymbol{a}^t$ at time $t$.

1: Object masks $\mathbf{O}^{t-h:t} \leftarrow$ ObjectDetector($\mathbf{I}^{t-h:t}$), $\mathbf{O}$ include dynamic and static masks $\mathbf{D}, \mathbf{S}$
2: Instance masks $\mathbf{X}^{t-h:t} \leftarrow$ InstanceLocalization($\mathbf{I}^{t-h:t}, \mathbf{D}^{t-h:t}$)
3: Predicted instance masks $\hat{\mathbf{X}}^{t+1} \leftarrow \varnothing$
4: **for** each instance mask $\boldsymbol{x}$ in $\mathbf{X}$ **do**
5:     Effects from spatial relations $\boldsymbol{m}_1^t \leftarrow$ RelationNet($\boldsymbol{x}^t, \mathbf{O}^t, \boldsymbol{a}^t$)
6:     Effects from temporal relations $\boldsymbol{m}_2^t \leftarrow$ InertiaNet($\boldsymbol{x}^{t-h:t}, \boldsymbol{a}^t$)
7:     Total effects $\boldsymbol{m}^t \leftarrow \boldsymbol{m}_1^t + \boldsymbol{m}_2^t$
8:     Predicted instance mask $\hat{\boldsymbol{x}}^{t+1} \leftarrow$ Transformation($\boldsymbol{x}^t, \boldsymbol{m}^t$)
9:     $\hat{\mathbf{X}}^{t+1} \leftarrow \hat{\mathbf{X}}^{t+1} \bigcup \hat{\boldsymbol{x}}^{t+1}$
10: **end for**
11: Background image $\mathbf{B}^{t+1} \leftarrow$ BackgroundConstructor($\mathbf{I}^t, \mathbf{S}^t$)
12: Predicted next frame $\hat{\mathbf{I}}^{t+1} \leftarrow$ Merge($\hat{\mathbf{X}}^{t+1}, \mathbf{B}^{t+1}$)

---

**Object Detector and Instance Localization Module.** Object Detector is a CNN module aiming to learn object masks from input image. An object mask describes the spatial distribution of a class of objects, which forms the fundamental building block of our object-oriented framework. Considering that instances of the same class are likely to have different motions, we append an Instance Localization Module to Object Detector to localize each dynamic instance to support instance-level dynamics learning. The class-specific object masks in conjunction with instance localization build the bridge to connect visual perception (Object Detector) with dynamics learning (Dynamics Net), which allows learning objects based on both appearances and dynamics.

Specifically, Object Detector receives image $\mathbf{I}^t \in \mathbb{R}^{H \times W \times 3}$ at timestep $t$ and then outputs object masks $\mathbf{O}^t \in [0, 1]^{H \times W \times n_O}$, including dynamic object masks $\mathbf{D}^t \in [0, 1]^{H \times W \times n_D}$ and static object masks $\mathbf{S}^t \in [0, 1]^{H \times W \times n_S}$, where $H$ and $W$ denote the height and width of the input image, $n_D$ and $n_S$ denotes the maximum class number of dynamic and static objects respectively, and $n_O = n_D + n_S$. The entry $\mathbf{O}_{u,v,i}$ indicates the probability that the pixel $\mathbf{I}_{u,v,:}$ belongs to the $i$-th object class. Then, Instance Localization Module uses the dynamic object masks to compute each single instance mask $\mathbf{X}_{:,:,i}^t \in [0, 1]^{H_M \times W_M} (1 \leq i \leq n_M)$, where $H_M$ and $W_M$ denote the

height and width of the bounding box of this instance and $n_M$ denotes the maximum number of localized instances. As shown in Figure 1, Instance Localization Module first samples a number of bounding boxes on the dynamic object masks and then select the regions, each of which contains only one dynamic instance. As we focus on the motion of rigid objects, the affine transformation is approximatively consistent for all pixels of each dynamic instance mask. Inspired by this, we define a discrepancy loss $\mathcal{L}_{\text{instance}}$ for a sampled region that measures the motion consistence of its pixels and use it as a selection score for selecting instance masks. To compute this loss, we first compute an average rigid transformation of a sampled region between two time steps based on the instance masks and masked image in this region, then apply this transformation to this region at the previous time step, and finally compared this predicted region with the region at the current time. Obviously, when a sampled region contains exactly one dynamic instance, this loss will be extremely small, and even zero when the object masks are perfectly learned. More details of the region proposal sampling and instance mask selection can be found in Appendix A.

**Dynamics Net.** Dynamics Net is designed to learn instance-based motion effects of actions, object-to-object spatial relations (Relation Net) and temporal relations of spatial states (Inertia Net), and then reason about the motion of each dynamic instance based on these effects. Its architecture is illustrated in Figure 2, which has an Effect Net for each class of objects. An Effect Net consists of one Inertia Net and several Relation Nets. As shown in the left subfigure, instance-level dynamics learning is performed, which means the motion of each dynamic instance is individually computed. We take as an example the computation of the motion of the $i$-th instance $\mathbf{X}^t_{:,:,i}$ to show the detailed structure of the Effect Net.

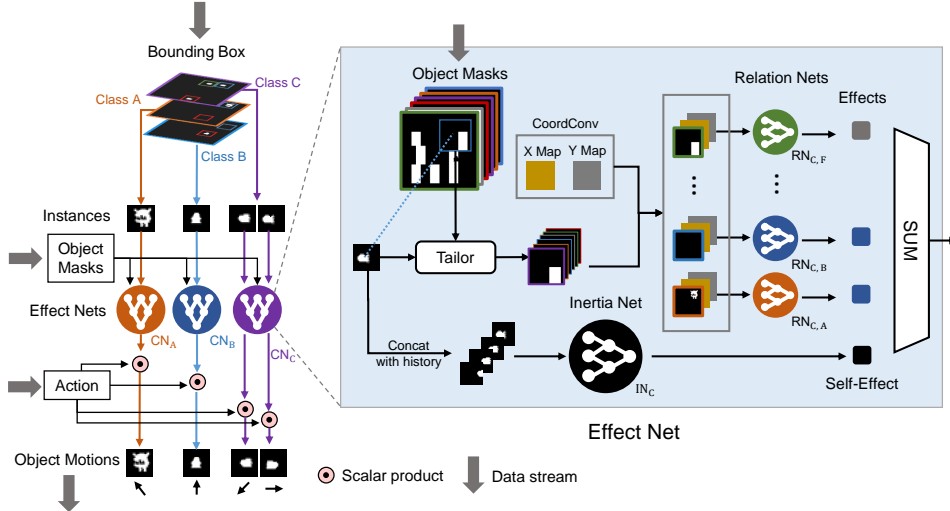

Figure 2: Architecture of Dynamics Net (left) and its component of Effect Net (right). Different classes are distinguished by different letters (e.g., A, B, ... , F).

As shown in the right subfigure of Figure 2, we first use a sub-differentiable tailor module introduced by Zhu et al. (2018) to enable the inference of dynamics focusing on the relations with neighbour objects. This module crops a $w$-size "horizon" window from the concatenated masks of all objects $\mathbf{O}^t$ centered on the expected location of $\mathbf{X}^t_{:,:,i}$, where $w$ denotes the maximum effective range of relations. Then, the cropped object masks are respectively concatenated with the constant x-coordinate and y-coordinate meshgrid map (to make networks more sensitive to the spatial information) and fed into the corresponding Relation Nets (RN) according to their classes. We use $\mathbf{C}^t_{:,:,i,j}$ to denote the cropped mask that crops the $j$-th object class $\mathbf{O}^t_{:,:,j}$ centered on the expected location of the $i$-th dynamic instance (the class it belongs to is denoted as $c_i$, $1 \leq c_i \leq n_D$). The effect of object class $j$ on class $c_i$, $E^t(c_i, j) \in \mathbb{R}^{2 \times n_a}$ ($n_a$ denotes the number of actions) is calculated as,

$$E^t(c_i, j) = \text{RN}_{c_i, j}\Big(\text{concat}\big(\mathbf{C}^t_{:,:,i,j}, \text{Xmap}, \text{Ymap}\big)\Big).$$ Note that there are total $n_D \times n_O$ RNs for $n_D \times n_O$ pairs of object classes that share the same architecture but not their weights. To handle the partial observation problem, we add an Inertia Nets (IN) to learn the self-effects of an object class

through historical states, $E_{\text{self}}^t(c_i) = \text{IN}_{c_i}\Big(\text{concat}\big(\mathbf{X}_{:,:,i}^t, \mathbf{X}_{:,:,i}^{t+1}, \ldots, \mathbf{X}_{:,:,i}^{t+h}\big)\Big)$, where $h$ is the history length and there are total $n_D$ INs for $n_D$ dynamic object classes that share the same architecture but not their weights. To predict the motion vector $\boldsymbol{m}_i^t \in \mathbb{R}^2$ for the $i$-th dynamic instance, all these effects are summed up and then multiplied by the one-hot coding of action $\boldsymbol{a}^t \in \{0,1\}^{n_a}$, that is,

$$\boldsymbol{m}_i^t = \Big(\big(\textstyle\sum_j E^t(c_i, j)\big) + E_{\text{self}}^t(c_i)\Big) \cdot \boldsymbol{a}^t.$$

**Background Extractor.** This module extracts the static background of input image based on the static object masks learned by Object Detector and then it is combined with the predicted dynamic instances to predict the next visual observation. As Object Detector can decompose its observation into objects in an unseen environment with a different object layout, Background Constructor is able to generate a corresponding static background and support the visual observation prediction in novel environments. Specifically, Background Constructor maintains an external background memory $\mathbf{B} \in \mathbb{R}^{H \times W \times 3}$ which is continuously updated (via moving average) by the static object mask learned by Object Detector. Denoting $\alpha$ as the decay rate, the updating formula is given by, $\mathbf{B}^t = \alpha\mathbf{B}^{t-1} + (1 - \alpha)\mathbf{I}^t \sum_i \mathbf{S}_{:,:,i}^t$, $\mathbf{B}^0 = 0$.

**Prediction and Training Loss.** At the output end of our model, the prediction of the next frame is produced by merging the learned object motions and the background $\mathbf{B}^t$. The pixels of a dynamic instance can be calculated by masking the raw image with the corresponding instance mask and we can use Spatial Transformer Network (STN) (Jaderberg et al., 2015) to apply the learned instance motion vector $\boldsymbol{m}_i^t$ on these pixels. First, we transform all the dynamic instances according to the learned instance-level motions. Then, we merge all the transformed dynamic instances with the background image calculated from Background Extractor to generate the prediction of the next frame. In this paper, we use the pixel-wise $l_2$ loss to restrain image prediction error, denoted as $\mathcal{L}_{\text{prediction}}$. To get earlier feedback before reconstructing images and facilitate the training process, we add a highway loss, $\mathcal{L}_{\text{highway}} = \sum_i \big\| (\bar{u}_i, \bar{v}_i)^t + \boldsymbol{m}_i^t - (\bar{u}_i, \bar{v}_i)^{t+1} \big\|_2^2$, where $(\bar{u}_i, \bar{v}_i)^t$ is the excepted location of $i$-th instance mask $\mathbf{X}_{:,:,i}^t$. In addition, we add a proposal loss to utilize the dynamic instance proposals provided by the abstracted problem to guide our optimization, which is given by $\mathcal{L}_{\text{proposal}} = \big\| \sum_i (\mathbf{D}_{:,:,i}^t - \mathbf{P}_{:,:,i}^t) \big\|_2^2$, where $\mathbf{P}$ denotes the dynamic instance region proposals computed by the more abstract learning level (i.e., dynamic instance segmentation level). The total loss of the dynamics learning level is given by combining the previous losses with different weights,

$$\mathcal{L}_{\text{DL}} = \mathcal{L}_{\text{highway}} + \lambda_1 \mathcal{L}_{\text{prediction}} + \lambda_2 \mathcal{L}_{\text{proposal}}$$

## 3.2 DYNAMIC INSTANCE SEGMENTATION LEVEL

This level aims to generate region proposals of dynamic instances to guide the learning of object masks and facilitate instance localization at the level of dynamics learning. The architecture is shown in Figure 1. Instance Splitter aims to identify regions, each of which potentially contains one dynamic instance. To learn to divide different dynamic object instances onto different masks, we use the discrepancy loss $\mathcal{L}_{\text{instance}}$ described in Section 3.1 to train Instance Splitter. Considering that one object instance may be split into smaller patches on different masks, we append a Merging Net (i.e., a two-layer CNN with 1 kernel size and 1 stride) to Instance Splitter to learn to merge redundant masks by a merging loss $\mathcal{L}_{\text{merge}}$ based on the prior that the patches of the same instance are adjacent to each other and share the same motion. In addition, we add a foreground proposal loss $\mathcal{L}_{\text{forground}}$ to encourage attentions on the dynamic regions. The total loss of this level is given by combining these losses with different weights,

$$\mathcal{L}_{\text{DIS}} = \mathcal{L}_{\text{instance}} + \lambda_3 \mathcal{L}_{\text{merge}} + \lambda_4 \mathcal{L}_{\text{forground}}$$

For more complex domains with arbitrary deformation and appearance change, MAOP is also readily to incorporate the vanilla Mask R-CNN (He et al., 2017) or other off-the-shelf supervised object detection methods (Liu et al., 2018) as a plug-and-play module into our framework to generate region proposals of dynamic instances. In addition, although the network structure of this level is similar to Object Detector in the level of dynamics learning, we do not integrated them together as a whole network because the concurrent learning of both object representations and dynamics model is not stable. Instead, we first learn the coarse object instances only based on the spatial-temporal consistency of locomotion and appearance pattern, and then use them as proposal regions to perform object-oriented dynamics learning at the more fine-grained level, which in turn fine-tunes the object representations.

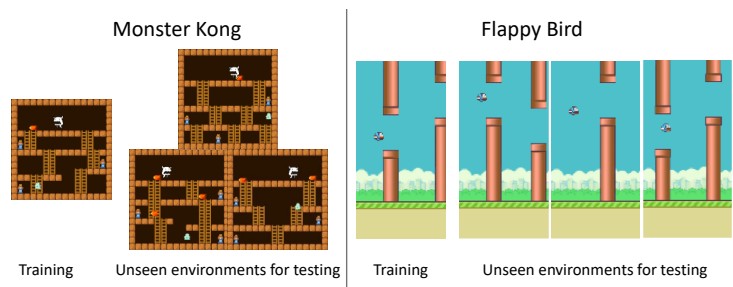

Figure 3: Examples of 1-to-3 generalization experiments.

### 3.3 MOTION DETECTION LEVEL

At this level, we employ foreground detection to detect changing regions from a sequence of image frames and provide coarse dynamic region proposals $\mathbf{F}_p$ for assisting in dynamic instance segmentation. In our experiments, we use a simple unsupervised foreground detection approach proposed by Lo & Velastin (2001). Our framework is also compatible with many advanced unsupervised foreground detection methods (Lee, 2005; Maddalena et al., 2008; Zhou et al., 2013; Guo et al., 2014) that are more efficient or more robust to moving camera. These complex unsupervised foreground detection methods have the potential to improve the performance but are not the focus of this work.

## 4 EXPERIMENTS

We evaluate our model on two games, *Monster Kong* and *Flappy Bird*, from the Pygame Learning Environment (Tasfi, 2016), which allows us to test generalization ability over various scenes with different layouts. Here, the *Monster Kong* is the advanced version of that used by Zhu et al. (2018), which has a more general and complex setting. The monster wanders around and breathes out fires randomly, and the fires also move with some randomness. The agent explores with actions *up*, *down*, *left*, *right*, *jump* and *noop*. All these dynamic objects interact with the environments and objects according to their own physics engine. Moreover, gravity and jump model has a long-term dynamics effects, leading to a partial observation problem. To test whether our model can truly learn the underlying physical mechanism behind the visual observations and perform relational reasoning, we set the $k$-to-$m$ zero-shot generalization experiment, where we use $k$ different environments for training and $m$ different unseen environments for testing. *Flappy Bird* is a side-scroller game, where a bird flies between columns of green pipes with action *jump* and *noop*. Since the unseen environments will be similar with the training ones without limitation of samples in this game, we limit the samples for training. Two experimental settings are shown in Figure 3. We use random exploration on *Monster Kong*, and an expert guided random exploration on *Flappy Bird* because in this domain a totally random exploration will lead to an early death of the agent even at the very beginning.

We compare MAOP with state-of-the-art action-conditioned dynamics learning baselines, AC Model (Oh et al., 2015), CDNA (Finn et al., 2016), and OODP (Zhu et al., 2018). AC Model adopts an encoder-LSTM-decoder structure, which performs transformations in hidden space and constructs pixel predictions. CDNA explicitly models pixel motions to achieve invariance to appearance. OODP and MAOP both aim at learning object-level dynamics through an object-oriented learning paradigm, which decomposes raw images into objects and perform prediction based on object-level relations. OODP is only designed for class-level dynamics, while MAOP is able to learn instance-level dynamics. Implementation details of MAOP can be found in Appendix C.

### 4.1 ZERO-SHOT GENERALIZATION AND SAMPLE EFFICIENCY

To make a sufficient comparison with the previous methods on the generalization ability and sample efficiency of object dynamics learning and image prediction, we conduct 1-5, 2-5 and 3-5 generalization experiments with a variety of evaluation indices on *Monster Kong*. We use $n$-error accuracy to measure the performance of object dynamics prediction, which is defined as the proportion that the difference between the predicted and ground-true agent locations is less than $n$ pixel. We

also add an extra pixel-based measurement (denoted by object RMSE), which compares the pixel difference near dynamic objects between the predicted and ground-truth images. To evaluate the image prediction, we adopt a typical image prediction loss (pixel-wise RMSE). In Figure 4 and B7 (Appendix B), we plot the learning curve for better visualization of the comparison in unseen environments. Further, we add a video (`https://github.com/maop2018/maop-video/blob/master/video.avi`) for better perceptual understanding of the prediction performance in unseen environments.

| | Models | Training environments | | | | | | | | Unseen environments | | | | | | | |
|---|---|---|---|---|---|---|---|---|---|---|---|---|---|---|---|---|---|
| | | 1-5† | | 1-5 | | 2-5 | | 3-5 | | 1-5† | | 1-5 | | 2-5 | | 3-5 | |
| | | Agent | All | Agent | All | Agent | All | Agent | All | Agent | All | Agent | All | Agent | All | Agent | All |
| 0-error accuracy | MAOP | 0.67 | 0.80 | 0.88 | 0.87 | 0.86 | 0.87 | 0.80 | 0.83 | **0.60** | **0.77** | **0.81** | **0.84** | **0.85** | **0.87** | **0.80** | **0.85** |
| | OODP | 0.24 | 0.17 | 0.18 | 0.16 | 0.22 | 0.17 | 0.26 | 0.20 | 0.20 | 0.16 | 0.20 | 0.15 | 0.18 | 0.15 | 0.26 | 0.18 |
| | AC Model | 0.04 | 0.59 | 0.87 | 0.94 | 0.80 | 0.93 | 0.77 | 0.92 | 0.01 | 0.18 | 0.08 | 0.16 | 0.30 | 0.48 | 0.45 | 0.66 |
| | CDNA | 0.30 | 0.66 | 0.41 | 0.76 | 0.42 | 0.78 | 0.44 | 0.74 | 0.31 | 0.55 | 0.37 | 0.59 | 0.40 | 0.71 | 0.41 | 0.70 |
| 1-error accuracy | MAOP | 0.90 | 0.91 | 0.97 | 0.94 | 0.97 | 0.93 | 0.96 | 0.93 | **0.86** | **0.90** | **0.96** | **0.93** | **0.97** | **0.93** | **0.95** | **0.93** |
| | OODP | 0.49 | 0.29 | 0.32 | 0.23 | 0.34 | 0.23 | 0.35 | 0.25 | 0.39 | 0.25 | 0.34 | 0.22 | 0.32 | 0.21 | 0.34 | 0.22 |
| | AC Model | 0.07 | 0.63 | 0.98 | 0.99 | 0.95 | 0.98 | 0.94 | 0.98 | 0.02 | 0.34 | 0.15 | 0.26 | 0.52 | 0.67 | 0.66 | 0.77 |
| | CDNA | 0.42 | 0.84 | 0.48 | 0.86 | 0.48 | 0.86 | 0.51 | 0.87 | 0.45 | 0.82 | 0.45 | 0.83 | 0.47 | 0.84 | 0.48 | 0.86 |
| 2-error accuracy | MAOP | 0.95 | 0.94 | 0.99 | 0.96 | 0.99 | 0.95 | 0.98 | 0.94 | **0.95** | **0.94** | **0.98** | **0.95** | **0.99** | **0.95** | **0.98** | **0.95** |
| | OODP | 0.67 | 0.47 | 0.44 | 0.37 | 0.46 | 0.32 | 0.49 | 0.39 | 0.60 | 0.43 | 0.48 | 0.34 | 0.43 | 0.31 | 0.46 | 0.36 |
| | AC Model | 0.10 | 0.64 | 0.99 | 0.99 | 0.98 | 0.99 | 0.97 | 0.99 | 0.04 | 0.34 | 0.20 | 0.31 | 0.64 | 0.73 | 0.77 | 0.81 |
| | CDNA | 0.50 | 0.86 | 0.52 | 0.87 | 0.53 | 0.88 | 0.54 | 0.88 | 0.53 | 0.85 | 0.47 | 0.84 | 0.50 | 0.86 | 0.51 | 0.87 |
| Object RMSE | MAOP | 31.99 | | 26.65 | | 31.68 | | 30.33 | | **34.14** | | **29.78** | | **31.32** | | **30.80** | |
| | OODP | 65.51 | | 66.44 | | 66.66 | | 64.73 | | 67.39 | | 67.41 | | 67.78 | | 64.95 | |
| | AC Model | 62.02 | | 18.88 | | 22.39 | | 21.30 | | 85.46 | | 57.41 | | 55.45 | | 43.48 | |
| | CDNA | 53.89 | | 34.99 | | 35.26 | | 35.94 | | 56.31 | | 45.34 | | 37.59 | | 37.80 | |
| Image RMSE | MAOP | 6.90 | | 5.64 | | 6.68 | | 6.46 | | **7.90** | | **8.60** | | 8.73 | | **6.55** | |
| | OODP | 14.70 | | 15.08 | | 14.89 | | 14.42 | | 15.42 | | 24.68 | | 26.39 | | 14.52 | |
| | AC Model | 15.99 | | 4.12 | | 4.78 | | 4.69 | | 44.92 | | 39.46 | | 38.07 | | 38.12 | |
| | CDNA | 11.47 | | 7.41 | | 7.58 | | 7.68 | | 12.23 | | 9.87 | | **8.10** | | 8.16 | |

Table 1: Prediction performance on *Monster Kong*. $k$-$m$ means the $k$-to-$m$ generalization problem. † indicates training with only 1000 samples. ALL represents all dynamic objects.

As shown in Table 1, MAOP significantly outperforms other methods in all experiment settings in terms of generalization ability and sample efficiency of both object dynamics learning and image prediction. It can achieve $0.84$ 0-error accuracy, even with a single training environment, which suggests MAOP is good at relational reasoning. Although AC Model achieves high accuracy in training environments, its performance in unseen scenes is much worse, which is probably because its pure pixel-level inference easily leads to overfitting. CDNA performs better than AC Model in those uncontrolled objects, but still cannot deal with complicated interactions in lack of knowledge on object-to-object relations. By the structural limitation of OODP, it has innate difficulty on frames with multiple dynamic objects.

| | Model | Training environments | | | | | | Unseen environments | | | | | |
|---|---|---|---|---|---|---|---|---|---|---|---|---|---|
| | | 0-acc | | 1-acc | | 2-acc | | 0-acc | | 1-acc | | 2-acc | |
| | | Agent | All | Agent | All | Agent | All | Agent | All | Agent | All | Agent | All |
| 3-steps | MAOP | 0.52 | 0.49 | 0.79 | 0.76 | 0.88 | 0.82 | **0.51** | 0.48 | **0.77** | 0.77 | **0.88** | **0.83** |
| | OODP | 0.05 | 0.08 | 0.15 | 0.12 | 0.27 | 0.16 | 0.04 | 0.07 | 0.15 | 0.12 | 0.26 | 0.16 |
| | AC Model | 0.48 | 0.77 | 0.68 | 0.88 | 0.80 | 0.92 | 0.18 | 0.45 | 0.35 | 0.66 | 0.48 | 0.72 |
| | CDNA | 0.15 | 0.74 | 0.20 | 0.77 | 0.22 | 0.78 | 0.21 | **0.74** | 0.26 | **0.78** | 0.28 | 0.78 |
| 8-steps | MAOP | 0.25 | 0.17 | 0.52 | 0.25 | 0.64 | 0.28 | **0.24** | 0.17 | **0.49** | 0.28 | **0.62** | 0.33 |
| | OODP | 0.03 | 0.08 | 0.05 | 0.10 | 0.08 | 0.11 | 0.02 | 0.10 | 0.05 | 0.11 | 0.09 | 0.13 |
| | AC Model | 0.12 | 0.11 | 0.18 | 0.14 | 0.23 | 0.16 | 0.00 | 0.03 | 0.01 | 0.04 | 0.02 | 0.05 |
| | CDNA | 0.16 | 0.68 | 0.24 | 0.72 | 0.29 | 0.75 | 0.16 | **0.68** | 0.24 | **0.72** | 0.29 | **0.75** |

Table 2: Performance of long-range prediction on *Monster Kong*. $n$-acc means $n$-error accuracy. MAOP and OODP are trained for 1-step prediction, while AC Model and CNDA are trained for 3-step prediction. They are all tested for 3-step and 8-step prediction.

In addition, we evaluate our performance of long-range prediction, as shown in Table 2. For action-conditioned dynamics prediction, our model only trained for 1-step prediction can achieve better performance than AC Model and CDNA trained for 3-step prediction. Specifically, our model has

88% probability that the predicted and ground-true agent locations are below 2 pixel when predicting 3 steps of the future, while the probability is 62% when predicting 8 steps of the future, which is also a satisfactory performance. Because the uncontrolled background objects (e.g., fires) tend to move according to a certain pattern and LSTM is good at remembering specific patterns of dynamics, CDNA has the highest prediction performance for background objects but fails to predict the complex dynamics that is conditioned on actions and object relations. Figure B8 illustrates a case to visualize the 8-step prediction of our model in unseen environments.

We also test our model on *Flappy Bird*, where we limit the training samples to 100 and 300 to form a sufficiently challenging generalization task. As shown in Table 3, our performance is similar with that on *Monster Kong*. Our generalization ability and sample efficiency significantly outperform other baselines. Surprisingly, only 100 samples are enough to reach almost perfect 1-error accuracy.

| | Models | Training environments | | | | Unseen environments | | | |
| --- | --- | --- | --- | --- | --- | --- | --- | --- | --- |
| | | 1-5† | | 1-5‡ | | 1-5† | | 1-5‡ | |
| | | Agent | All | Agent | All | Agent | All | Agent | All |
| 0-error accuracy | MAOP | **0.84** | **0.90** | **0.87** | **0.93** | **0.83** | **0.89** | **0.83** | **0.92** |
| | OODP | 0.01 | 0.29 | 0.01 | 0.32 | 0.01 | 0.18 | 0.02 | 0.15 |
| | AC Model | 0.39 | 0.64 | 0.48 | 0.75 | 0.03 | 0.18 | 0.04 | 0.23 |
| | CDNA | 0.13 | 0.78 | 0.41 | 0.84 | 0.10 | 0.77 | 0.16 | 0.79 |
| 1-error accuracy | MAOP | **0.99** | **1.00** | **0.97** | **0.97** | **0.99** | **0.99** | **0.98** | **0.97** |
| | OODP | 0.05 | 0.52 | 0.04 | 0.56 | 0.06 | 0.39 | 0.07 | 0.39 |
| | AC Model | 0.48 | 0.80 | 0.57 | 0.87 | 0.07 | 0.37 | 0.14 | 0.45 |
| | CDNA | 0.26 | 0.82 | 0.57 | 0.89 | 0.22 | 0.81 | 0.36 | 0.84 |
| 2-error accuracy | MAOP | **1.00** | **1.00** | **0.99** | **0.99** | **1.00** | **1.00** | **0.99** | **0.98** |
| | OODP | 0.14 | 0.66 | 0.12 | 0.67 | 0.16 | 0.59 | 0.16 | 0.56 |
| | AC Model | 0.53 | 0.85 | 0.63 | 0.90 | 0.12 | 0.53 | 0.24 | 0.64 |
| | CDNA | 0.37 | 0.84 | 0.66 | 0.92 | 0.36 | 0.84 | 0.49 | 0.87 |

Table 3: Performance of the object dynamics prediction on 1-5 generalization problem on *Flappy Bird*. † and ‡ indicates training with only 100 and 300 samples.

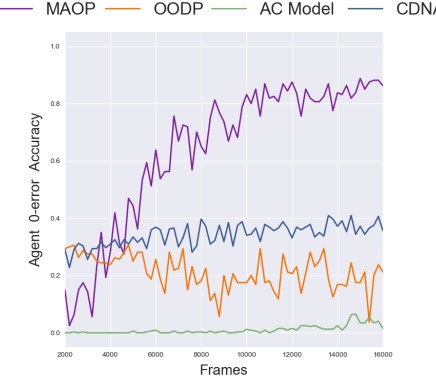



Figure 4: Learning curves for the dynamics prediction of the agent in unseen environments on *Monster Kong*.

Figure 5: The ground-truth label distribution of our discovered controllable agents in unseen environments on *Monster Kong*.

## 4.2 INTERPRETABLE REPRESENTATIONS AND KNOWLEDGE

MAOP takes a step towards interpretable deep learning and disentangled representation learning. Through interacting with environments, it learns fruitful visually and semantically interpretable knowledge in an unsupervised manner, which contributes to unlock the "black box" of neural networks and open the avenue for further researches on object-based planning, object-oriented model-based RL, and hierarchical learning.

**Visual interpretability.** To demonstrate the visual interpretability of MAOP in unseen environments, we visualize the learned masks of dynamic and static objects. We highlight the attentions of

the object masks by multiplying the raw images by the binarized masks. Note that MAOP does not require the actual number of objects but a maximum number and some learned object masks may be redundant. Thus, we only show the informative object masks. As shown in Figure 6, our model captures all the key objects in the environments including the controllable agents (the cowboy and the bird), the uncontrollable dynamic objects (the monster, fires and pipes), and the static objects that have effects on the motions of dynamic objects (ladders, walls and the free space). We also observe that model can learn disentangled object representations and distinguish the objects by both appearance and dynamic property.

Monster Kong                    Flappy Bird

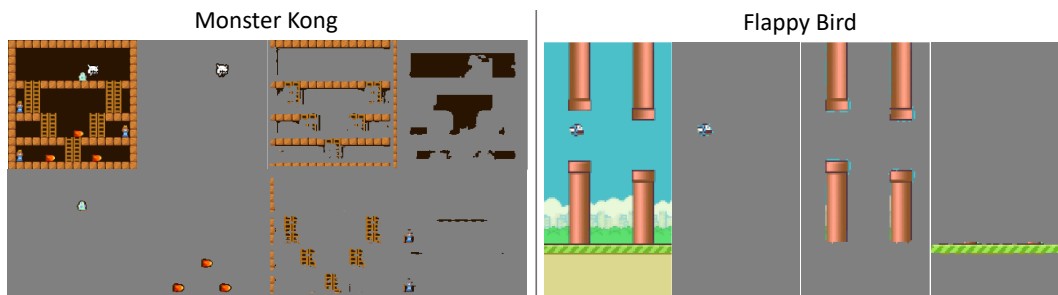

Figure 6: Visualization of the masked images in unseen environments. Left is the raw image.

**Discovery of the controllable agent.** With the learned knowledge in MAOP, we can easily uncover the action-controlled agent from all the dynamic objects, which is useful semantic information that can be used in heuristic algorithms. Specifically, the object that has the maximal variance of total effects over actions is the action-controlled agent. Denote the total effects as $E_i = (\sum_j E(c_i, j)) + E_{\text{self}}(c_i), E_i \in \mathbb{R}^{2 \times n_a}$, the label of the action-controlled agent is calculated as, $\arg\max_i Var(E_i)$. The histogram in Figure 5 plotting the ground-truth label distribution of our discovered action-controlled agents clearly demonstrates that our discovery of the controllable agent achieves perfect 100% accuracy.

**Dynamical interpretability.** To show the dynamical interpretability behind image prediction, we test our predicted motions by comparing RMSEs between the predicted and ground-truth motions in unseen environments (Table B4 in Appendix B). Intriguingly, most predicted motions are quite accurate, with the RMSEs less than 1 pixel. Such a visually indistinguishable error also verifies our dynamics learning.

## 5 CONCLUSIONS AND FUTURE WORK

This paper presents a self-supervised multi-level learning framework for learning action-conditioned object-based dynamics. This framework is sample-efficient and generalizes object dynamics and prediction of raw visual observations to complex unseen environments with multiple dynamic objects. The learned dynamics model potentially enables an agent to directly plan or efficiently learn for unseen environments. Although a random policy or an expert's policy is used for exploration in our experiments, our framework can support smarter exploration strategies, e.g., curiosity-driven exploration. Our future work includes extending our model for deformation prediction (e.g., object appearing, disappearing and non-rigid deformation) and incorporating a camera motion prediction network module introduced by Vijayanarasimhan et al. (2017) for applications such as FPS games and autonomous driving.

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

## APPENDIX A    INSTANCE LOCALIZATION

Instance localization is a common technique in context of supervised region-based object detection (Girshick et al., 2014; Girshick, 2015; Ren et al., 2015; He et al., 2017; Liu et al., 2018), which localizes objects on raw images with regression between the predicted bounding box and the ground truth. Here, we propose an unsupervised approach to perform dynamic instance localization on dynamic object masks learned by Object Detector. Our objective is to sample a number of region proposals on the dynamic object masks and then select the regions, each of which has exactly one dynamic instance. In the rest of this section, we will describe these two steps in details.

**Region proposal sampling.** We design a learning-free sampling algorithm for sampling region proposals on object masks. This algorithm generates multi-scale region proposals with a full coverage over the input mask. Actually, we adopt multi-fold full coverage to ensure that pixels of the potential instances are covered at each scale. The detailed algorithm is described in Algorithm 2.

**Instance mask selection.** Instance mask selection aims at selecting the regions, each of which contains exactly one dynamic instance, based on the discrepancy loss $\mathcal{L}_{\text{instance}}$ (Section 3.1). To screen out high-consistency, non-overlapping and non-empty instance masks at the same time, we integrate Non-Maximum Suppression (NMS) and Selective Search (SS) (Uijlings et al., 2013) in the context of region-based object detection (Girshick et al., 2014; Girshick, 2015; Ren et al., 2015; He et al., 2017; Liu et al., 2018) into our algorithm.

---

**Algorithm 2** Region proposal sampling.

---

**Input:** Dynamic object mask $\mathbf{D} \in [0,1]^{H \times W}$, the number of region proposal scales $n_S$, the folds of full coverage $T$ .

1: Initialize proposal set $\mathbb{P} = \varnothing$.
2: Binarize $\mathbf{D}$ to get the indicator for the existence of objects
3: **for** $l = 1 \ldots n_S$ **do**
4:     Select scale $dx, dy$ depend on the level $l$.
5:     **for** $t = 1 \ldots T$ **do**
6:         Initialize candidate set $\mathbb{C} = \{(i,j)|\mathbf{D}_{i,j} = 1\}$.
7:         **while** $\mathbb{C} \neq \varnothing$ **do**
8:             Sample a pixel coordinate $(x, y)$ from $\mathbb{C}$.
9:             Get a box $\mathbb{B} = \{(i,j)|\, |i - x| \leq dx, |j - y| \leq dy\}$.
10:            **if** $\mathbb{B}$ is not empty **then**
11:                Insert $\mathbb{B}$ into the proposal set $\mathbb{P} \leftarrow \mathbb{P} \cup \{\mathbb{B}\}$.
12:            **end if**
13:            Update the remain candidate set $\mathbb{C} \leftarrow \mathbb{C} \setminus \mathbb{B}$.
14:         **end while**
15:     **end for**
16: **end for**
17: **return** $\mathbb{P}$

---

## APPENDIX B    TABLES AND FIGURES

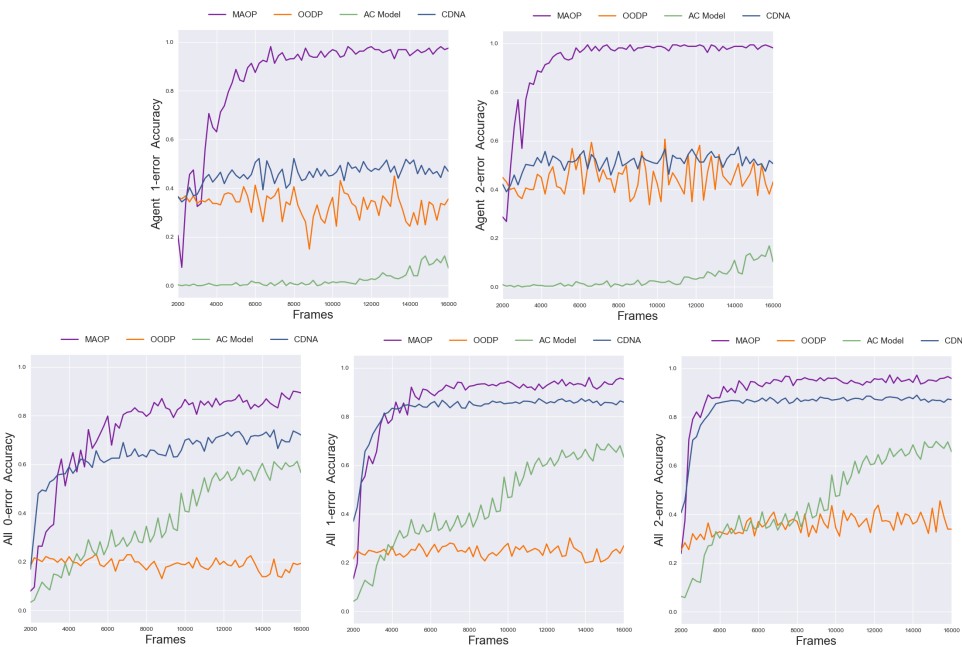

Figure B7: Learning curves for the dynamics prediction in unseen environments on *Monster Kong*. The curves with "Agent" notation represent the learning curves of the agent, while those with "All" notation indicate the learning curves of all dynamic objects.

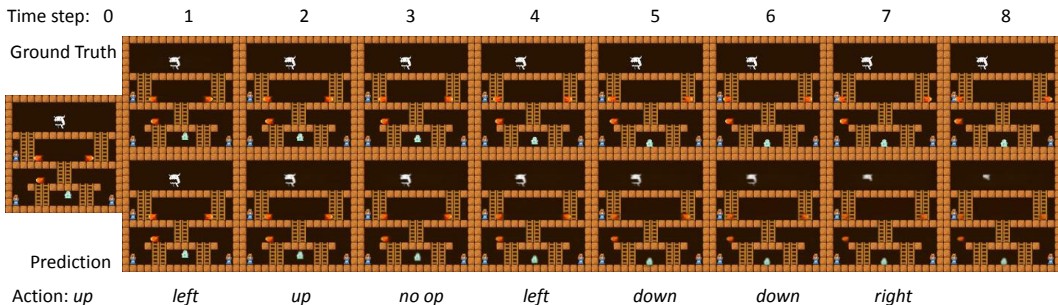

Figure B8: A case shows the 8-step prediction of our model in unseen environments on *Monster Kong*.

| Model | Monster Kong | | | | Flappy Bird | |
|---|---|---|---|---|---|---|
| | 1-5† | 1-5 | 2-5 | 3-5 | 1-5† | 1-5‡ |
| MAOP | 1.96 | 0.34 | 0.38 | 0.42 | 0.30 | 0.34 |

Table B4: Average motion prediction error in two experiment environments. † and ‡ correspond to the same sample restriction experiments in Table 1 and 3

## APPENDIX C    IMPLEMENTATION DETAILS FOR EXPERIMENTS

Object Detector in the dynamics learning level and Instance Splitter in the dynamic instance segmentation level have similar architectures with Object Detector in OODP (Zhu et al., 2018). To augment the interactions of instances when training Instance Splitter, we random sample two region proposals and combine them into a single region proposal with double size.

Denote $Conv(F, K, S)$ as the convolutional layer with the number of filters $F$, kernel size $K$ and stride $S$. Let $R(), S()$ and $BN()$ denote the ReLU layer, sigmoid layer and batch normalization layer (Ioffe & Szegedy, 2015). The 5 convolutional layers in Object Detector can be indicated as $R(BN(Conv(16, 5, 2)))$, $R(BN(Conv(32, 3, 2)))$, $R(BN(Conv(64, 3, 1)))$, $R(BN(Conv(32, 1, 1)))$, and $BN(Conv(1, 3, 1))$, respectively. The 5 convolutional layers in Instance Detector can be indicated as $R(BN(Conv(32, 5, 2)))$, $R(BN(Conv(32, 3, 2)))$, $R(BN(Conv(32, 3, 1)))$, $R(BN(Conv(32, 1, 1)))$, and $BN(Conv(1, 3, 1))$, respectively. The architecture of Foreground Detector is similar to binary-class Object Detector and the 5 convolutional layers in Foreground Detector can be indicated as $R(BN(Conv(32, 5, 2)))$, $R(BN(Conv(32, 3, 2)))$, $R(BN(Conv(32, 3, 1)))$, $R(BN(Conv(32, 1, 1)))$, and $S(BN(Conv(1, 3, 1)))$, respectively. The CNNs in Relation Net are connected in the order: $R(BN(Conv(16, 3, 2)))$, $R(BN(Conv(32, 3, 2)))$, $R(BN(Conv(32, 3, 2)))$, and $R(BN(Conv(32, 3, 2)))$. The last convolutional layer is reshaped and fully connected by the 64-dimensional hidden layer and the 2-dimensional output layer successively. Inertia Net has the same architecture and hyperparameters as Relation Net.

The hyperparameters for training MAOP in *Monster Kong* and *Flappy Bird* are listed as follows:

- The weights of losses, $\lambda_1$, $\lambda_2$, and $\lambda_3$, are 100, 1, and 10, respectively. In addition, all the $l_2$ losses are divided by $HW$ to keep invariance to the image size.
- Batch size is 16 and the maximum number of training steps is set to be $1 \times 10^5$.
- The optimizer is Adam (Kingma & Ba, 2014) with learning rate $1 \times 10^{-3}$.
- The raw images of *Monster Kong* and *Flappy Bird* are resized to $160 \times 160 \times 3$ and $160 \times 80 \times 3$ , respectively.
- Overlapping threshold $\alpha$ is 0.5.
- The size of the horizon window $w$ is 33 on *Monster Kong*, 41 on *Flappy Bird*.
- The maximum number of static and dynamic masks is 4 and 12 on *Monster Kong*, on *Flappy Bird*.
- The maximum instance number of each object class is set to be 15.

