# OpenReview forum: "Object-Oriented Model Learning through Multi-Level Abstraction"
_ICLR.cc/2019/Conference_

### Official Review · AnonReviewer3 · 2018-11-01
**Proposed model can betrained sucessfully on video game frames, but appears highly engineered and not very generic. Paper could be structured better to improve readibility**

**Rating:** 6
**Confidence:** 3

**Review:**

In this paper, the novel MAOP model is described for self-supervised learning on past video game frames to predict future frames. The presented results indicate that the method is capable of discovering semantically important visual components, and their relation and dynamics, in frames from arcade-style video games. Key to the approach is a multi-level self-learning approach:  more abstract stages focus on simpler problems that are easier to learn, which in turn guide the learning process at the more complex stages.
A downside is that it the method is complex, consisting of many specific sub-components and algorithms, which in turn have again other sub-components. This makes the paper a long read with a lot of repetition, and various times the paper refers to the names of sub-components that are only explained later. Other methodological details that are relevant to understand how the method operates are described in the Appendices. I expect that if the paper would be better structured, it would be easier to understanding how all the parts fit together. Another downside of this complexity is that the method seems designed for particular types of video game frames, with static backgrounds, a fixed set of objects or agents. It is unclear how the method would perform on other types of games, or on real-world videos. While the method therefore avoids the need for manual annotation, it instead encodes a lot of domain knowledge in its design and components.
I also didn't fully understand how the self-supervised model is used for Reinforcement Learning in the experiments. Is the MAOP first trained, and the fixed to perform RL with the learned agent models, or is the MOAP learned end-to-end during RL?

Pros:
+ MAOP seems successful on the tested games in the experiments
+ Demonstrates that, with a sufficiently engineered method, self-supervised learning can be used to discover different types of objects, and their dynamics.

Cons:
- writing could be improved, as the methodology currently reads as a summation of facts, and some parts are written out of order, resulting in various forward references to components that only become clear later. Several times, the paper states that some novel algorithm is used, but then provides no further explanation in the text as all description of this novelty is deferred to an appendix.
- method does not seem generic, hence it is unclear how relevant this architecture it is to other use cases
- many hyperparameters for the individual components, algorithms. Unclear how these parameter setting affect the results

Below are more detailed comments and questions:

General comments:
* The proposed MOAP method consists of many subalgorithms, resulting in various (hyper)parameters which may impact the results (e.g. see Appendix A, B). Appendix D lists several used hyperparameter settings, though various parameters for the algorithms are still missing (e.g. thresholds alpha, beta in Algo.2). Were the used parameters optimized? How are these hyperparameters set in practice? How does changing them impact your results?
* Methods seems particularly designed for 'video games', where the object and background structures have well defined sizes, appearance, etc. How will the MOAP fair in more realistic situations with noisy observations, occluded objects, changing appearances and lighting conditions, etc.?
* How about changing appearance of an agent during an action, e.g. a 'walking animation' ? Can your method learn the sequence of sprites to accurately predict the next image? Is that even part of the objective?
* Appendix D has important implementation details, but is never mentioned in the text I believe! Didn't realize it existed on first read through.

* Introduction:
	* What prediction horizon are you targeting? 1 step, T steps into the future, 1 to T steps in the future simultaneously?
	What are you trying to predict? Object motion? Future observations?
	* "... which includes a CNN-based Relation Net to ... ", the names Relation Net, Inertia Net, etc.. are used as if the reader is expected to know what these are already. If these networks were introduced in related work already, please add citations. Otherwise please rephrase to clarify that these are networks themselves are part of your novel design.

* Section 3.1
	* "It takes multiple-frame video images ... and produce the predictions of raw visual observations.". As I understand from this, the self-supervised approach basically performs supervised learning to predict a future frame (target output) given past frames (input). I do not understand how this relates to Reinforcement Learning (RL) as mentioned in the introduction and Related Work. Is there still some reward function in play when learning the MAOP parameters? Or is the idea to first self-supervised learn the MAOP, and afterwards fix its parameters and use it in separate a RL framework? I believe RL is not mentioned anymore until Section 4.2. This connection between self-supervised and reinforcement learning should be clarified, or otherwise the related work should be adjusted to include other (self-supervised) work on predicting future image frames.
	* "An object mask describes the spatial distribution of an object ..." Does the distribution capture uncertainty on the object's location, or does it capture the spread of the object's extent ('mass distribution') ?
	* "Note that Object Detector uses the same CNN architecture with OODP". What does OODP stand for? Add citation here. (first mention of OODP is in Experiments section)
	* "(similar with Section 3.2)" → "similar to". Also, I find it a confusing to say something is similar to what will be done in a future section, which has not yet been introduced. Can you not explain the procedure here, and in Section 3.2 say that the procedure is "similar to Section 3.1" instead?
	* "to show the detailed structure of the Effect Net module." First time I see the name 'Effect Net', what is it? This whole paragraph different nets are named, with a rough indication of their relation, such as "Dynamic Net", "Relation Net" and "Inertia Net". Is "Effect Net" a different name for any of the three previous nets? The paper requires the reader to puzzle from Fig.2 that Relation Net and Inertia Net are parts of Effect Net, which in turn is part of Dynamics Net. This wasn't clear from the text at all.

* Section 3.2:
	* p7.: "Since DISN leans" → "Since DISN learns" ?
	* There are many losses throughout the paper, but I only see at the end of Section 3.1 some mentioning that multiple losses are combined. How is this done for the other components, .e.g is the total loss for DISN a weighted sum of L_foreground and L_instance ? Are the losses for all three three MAOP levels weighted for full end-to-end learning?
	* This section states various times "we propose a novel [method]", for which then no explanation is given, and all details are explained in the Appendix. While the Appendix can hold important implementation details, I would still expect that novelties of the paper are clearly explained in the paper itself. As it stands, the appendix is used as an extension of the methodological section of an already lengthy paper.
	* "Conversely, the inverse function is ... " M has a mask for each of the n_o "object classes", hence the "Instance Localization Module" earlier to split out instances from the class masks. So how can there be a single motion vector STN^-1(M,M') if there are multiple instances for an object mask? How will STN^-1 deal with different amount of instances in M and M' ?

* Section 3.3:
	* What is the output of this level? I expect some mathematical formulation as in the previous sections, resulting in some symbol, that is then used in Section 3.2. E.g. is the output "foreground masks F" (found in Appendix A) ?  This paper is a bit of a puzzle through the pages for the reader.

* Section 4:
	* "We compare MAOP with state-of-the-art action-conditioned dynamics learning baselines, ..." Please re-iterate how these methods differ in assumptions, what they model, with respect to your novel method? For instance, is the main difference your "novel region proposal method" and such? Is the overall architecture different? E.g. explain here already the AC Model uses "pixel-level inference", and that OODP has "lacks knowledge on object-to-object relations" to underline their difference to your approach, and provide context for your conclusions in Section 4.1.

* Appendix A:
	* Algorithm 1, line 7: "sample a pixel coordinate" → is this non-deterministically sampling?

---

> ### Author Response · Authors · 2018-11-28
> **Response to Reviewer 3**
>
> Thank you for your thoughtful review and suggestions.
>
> Q: Another downside of this complexity is that the method seems designed for particular types of video game frames, with static backgrounds, a fixed set of objects or agents.
>
> A: Our methods are designed for general settings with rigid objects and without camera motion. We did not assume there are fixed set of objects and agents. The monster breathes out fires randomly, so the existence and the number of the fires are dyanmic. For dealing with a dynamic background with camera motion, it is possible to incorporate a camera motion prediction network module introduced by [1] for applications such as FPS games and autonomous driving.
>
> Q; * The proposed MOAP method consists of many subalgorithms, resulting in various (hyper)parameters which may impact the results (e.g. see Appendix A, B). Appendix D lists several used hyperparameter settings, though various parameters for the algorithms are still missing (e.g. thresholds alpha, beta in Algo.2). Were the used parameters optimized? How are these hyperparameters set in practice? How does changing them impact your results?
>
> A: Thresholds alpha, beta are parameters for eliminating empty and overlapping masks, which are also used in Non-Maximum Suppression (NMS) in the context of region-based object detection (e.g., [2]). We set 10 for alpha and 0.5 for beta in all games. Our model is not sensitive to these parameters because empty and overlapping masks only increase the computation and do not influence the accuracy.
>
> Q: * Methods seems particularly designed for 'video games', where the object and background structures have well defined sizes, appearance, etc. How will the MOAP fair in more realistic situations with noisy observations, occluded objects, changing appearances and lighting conditions, etc.?
> * How about changing appearance of an agent during an action, e.g. a 'walking animation' ? Can your method learn the sequence of sprites to accurately predict the next image? Is that even part of the objective?
>
> A: Our objective is to learn the locomotion dynamics of rigid objects. As MOAP uses the CNN modules for object detection and spatial-relation reasoning, it is robust to some changes to object appearances, which has been demonstrated by the baseline approach of OODP. As the large appearance changes and arbitrary deformations are often predictable from a long-term view or memory, our proposed architecture can incorporate memory networks (e.g., LSTM) to handle such changes, which is our future work.
>
> Q: * What prediction horizon are you targeting? 1 step, T steps into the future, 1 to T steps in the future simultaneously? What are you trying to predict? Object motion? Future observations?
>
> A: We are trying to predict both the object motion and the future observations for 1 to T steps in the future.
>
> Q: I also didn't fully understand how the self-supervised model is used for Reinforcement Learning in the experiments. Is the MAOP first trained, and the fixed to perform RL with the learned agent models, or is the MOAP learned end-to-end during RL?
>
> A: We can first train MAOP and then perform RL with the learned model. For example, our learned dynamics model can guide the exploration of model-free RL [3], or be used with existing policy search or planning methods (e.g., MCTS and MPC) [4], or directly plugged into an end-to-end policy network integrating model-free and model-based path [5]. MAOP can also be learned end-to-end during RL, like [6]. In addition, there are other ways to use MAOP in RL. For example, the prediction error of our predictive model can be used as signals for curiosity-driven exploration [7]. Our learned object representations can be leveraged to design effective heuristic reward functions (like the distance-based rewards [8]) to facilitate model-free RL, or used to set subgoals in hierarchical RL [9].
>
> Q: "An object mask describes the spatial distribution of an object ..." Does the distribution capture uncertainty on the object's location, or does it capture the spread of the object's extent ('mass distribution') ?
> A: It captures the spread of the object's extent. An object mask describes the spatial distribution of a class of objects. Each entry of one object mask represents the probability that the corresponding pixel belongs to this class of objects.

---

> ### Author Response · Authors · 2018-11-28
> **Response to Reviewer 3 (connected to the previous response)**
>
> Q: "to show the detailed structure of the Effect Net module." First time I see the name 'Effect Net', what is it? This whole paragraph different nets are named, with a rough indication of their relation, such as "Dynamic Net", "Relation Net" and "Inertia Net". Is "Effect Net" a different name for any of the three previous nets? The paper requires the reader to puzzle from Fig.2 that Relation Net and Inertia Net are parts of Effect Net, which in turn is part of Dynamics Net. This wasn't clear from the text at all.
>
> A: The Dynamics Net has different Effect Nets for different classes of objects. An Effect Net consists of one Inertia Net and one Relation Net.
>
> Q: Appendix A: * Algorithm 1, line 7: "sample a pixel coordinate" → is this non-deterministically sampling?
>
> A: Yes, it is random sampling.
>
> Q: "Conversely, the inverse function is ... " M has a mask for each of the n_o "object classes", hence the "Instance Localization Module" earlier to split out instances from the class masks. So how can there be a single motion vector STN^-1(M,M') if there are multiple instances for an object mask? How will STN^-1 deal with different amount of instances in M and M' ?
>
> A: Instance Localization Module first samples a number of bounding boxes on the dynamic object masks and then selects the regions, each of which contains only one dynamic instance. As we focus on the motion of rigid objects, the affine transformation is approximatively consistent for all pixels of each dynamic instance mask. Inspired by this, we define the discrepancy loss for a sampled region that measures the motion consistence of its pixels and use it as a selection score for selecting instance masks. To compute this loss, we first compute an average rigid transformation (i.e., STN^-1(M, M')) of a sampled region between two time steps based on the instance masks and masked image in this region, then apply this transformation to this region at the previous time step, and compared the predicted region from the previous time step with the region at the current time step. Obviously, when a sampled region contains exactly one dynamic instance, this loss will be extremely small, and even zero when the object masks are perfectly learned.
>
> Q: * "We compare MAOP with state-of-the-art action-conditioned dynamics learning baselines, ..." Please re-iterate how these methods differ in assumptions, what they model, with respect to your novel method? For instance, is the main difference your "novel region proposal method" and such? Is the overall architecture different? E.g. explain here already the AC Model uses "pixel-level inference", and that OODP has "lacks knowledge on object-to-object relations" to underline their difference to your approach, and provide context for your conclusions in Section 4.1.
>
> A: We describe how MAOP and baseline methods differ as follows and added these descriptions in Section 4. AC Model adopts an encoder-LSTM-decoder structure, which performs transformations in hidden space and constructs pixel predictions. CDNA explicitly models pixel motions to achieve invariance to appearance. OODP and MAOP both aim at learning object-level dynamics through an object-oriented learning paradigm, which decomposes raw images into objects and perform predictions based on object-level relations. OODP is only designed for class-level dynamics, while MAOP is able to learn instance-level dynamics.
>
>
> References:
> [1] Vijayanarasimhan, Sudheendra, et al. "Sfm-net: Learning of structure and motion from video." arXiv preprint arXiv:1704.07804 (2017).
> [2] Ren, Shaoqing, et al. "Faster r-cnn: Towards real-time object detection with region proposal networks." Advances in neural information processing systems. 2015.
> [3] Chiappa, Silvia, et al. "Recurrent environment simulators." arXiv preprint arXiv:1704.02254 (2017).
> [4] Finn, Chelsea, and Sergey Levine. "Deep visual foresight for planning robot motion." Robotics and Automation (ICRA), 2017 IEEE International Conference on. IEEE, 2017.
> [5] Racanière, Sébastien, et al. "Imagination-augmented agents for deep reinforcement learning." Advances in Neural Information Processing Systems. 2017.
> [6] Deisenroth, Marc Peter, Carl Edward Rasmussen, and Dieter Fox. "Learning to control a low-cost manipulator using data-efficient reinforcement learning." (2011): 57-64.
> [7] Pathak, Deepak, et al. "Curiosity-driven exploration by self-supervised prediction." International Conference on Machine Learning (ICML). Vol. 2017. 2017.
> [8] Srinivas, Aravind, et al. "Universal Planning Networks." arXiv preprint arXiv:1804.00645 (2018).
> [9] Kulkarni, Tejas D., et al. "Hierarchical deep reinforcement learning: Integrating temporal abstraction and intrinsic motivation." Advances in neural information processing systems. 2016.

---

### Official Review · AnonReviewer1 · 2018-11-02

**Rating:** 4
**Confidence:** 3

**Review:**

This paper proposes a novel architecture, coined Multi-Level Abstraction Object-Oriented Predictor, MAOP. This architeture is composed of 3 parts, a Dynamics model, an object segmentation model, and a motion detection module.

While some parts of the model use handcrafted algorithms to extract data (e.g. the motion detection), most parts are learned and can be trained without much additional supervision, as the objectives are mostly unsupervised objectives.

The proposed model is interesting, and certainly "solves" the two tasks it is trained on. On the other hand, this model seems to be specifically tailored to solve these two tasks. It assumes a static background, very local newtonian-like physics, a very strong notion of object and object class. It is not clear to me if any of the improvements seen in this paper are valuable, reusable methods, or just good engineering work.
As such, I do not think that this paper fits ICLR. There has been a growing number of works that aim to find learning algorithms that learn to discover and disentangle object-like representations without having so much prior put into the model, but rather through some general purpose objective. The current paper seems like a decent applications paper, but it explores improvements orthogonal to this trend that IMO is what preoccupies the ICLR audience.

The writing of this paper makes it a bit hard to understand what the novel contributions of this paper are, and how the proposed method should go beyond the two problems that it solves. In general, there are many phrasings that would benefit from being rewritten more concisely; it would help with clarity, since the proposed model has a multitude of different parts with sometimes long names.

Experimentally, there are many parts to the proposed model, and while it is clear what each of them achieves, it is unclear how necessary each of the parts are, and how sensitive the model is to any part being (possibly slightly) incorrect.

The proposed method is tested on, presumably, RL environments; yet, no RL experiments are performed, so there is no way of knowing if the proposed model is actually useful for planning (there are instances of model-based methods learning acceptable models that are just wrong enough to *not* be useful to actually do RL or e.g. MCTS planning).

Overall, this paper tackles its tasks in an interesting but maybe too specific way; in addition, it could be improved in a variety of ways, both in terms of presentation and content. While the work is novel, I am not convinced that it is relevant to the interests of the ICLR audience.


Comments:
- When running your experiments, do you report results averaged over multiple runs?
- Figure 4+C7: why does the x-axis start at 2000?
- I don't think Figure 5 is really necessary
- All figures: your captions could be improved by giving more information about what their figure presents. E.g. in Figure C7 I have no idea what the curves correspond to. Sure it's accuracy, but for which task? How many runs? Is it a running average? Etc.
- Where are the test curves? Or are all curves test curves?
- Your usage of \citep and \citet, (Author, year) vs Author (year), is often inconsistent with how the citation is used.

---

> ### Author Response · Authors · 2018-11-28
> **Response to Reviewer 1**
>
> Thank you for your review and suggestions.
>
> Q: While some parts of the model use handcrafted algorithms to extract data (e.g. the motion detection)
>
> A: For motion detection, we did not use handcrafted algorithms. Instead, as described in Section 3.3, we use a simple unsupervised foreground detection approach proposed by [1]. Our MAOP framework is also compatible with many advanced unsupervised foreground detection methods [2-5] that are more efficient or more robust to moving camera. These complex unsupervised foreground detection methods have the potential to improve the performance.
>
> Q: The proposed model is interesting, and certainly "solves" the two tasks it is trained on. On the other hand, this model seems to be specifically tailored to solve these two tasks. It assumes a static background, very local newtonian-like physics, a very strong notion of object and object class. It is not clear to me if any of the improvements seen in this paper are valuable, reusable methods, or just good engineering work.
>
> A: We have addressed these concerns in the general response to all reviewers. The assumption of our method is that the environment only contains rigid objects and has no camera motion, which has been adopted by many papers (e.g., [6,7]). For dynamic background with camera motion, it is possible to incorporate a camera motion prediction network module introduced by [8] for applications such as FPS games and autonomous driving.
>
> Q: As such, I do not think that this paper fits ICLR. There has been a growing number of works that aim to find learning algorithms that learn to discover and disentangle object-like representations without having so much prior put into the model, but rather through some general-purpose objective. The current paper seems like a decent applications paper, but it explores improvements orthogonal to this trend that IMO is what preoccupies the ICLR audience.
>
> A: This paper aims to concurrently learn object-like representation and its action-conditioned dynamics in a self-supervised manner. We do not think our approach has over-built in prior knowledge. The main priors we used is that objects are represented by distribution masks and their dynamics are conditioned on actions and their spatial-temporal relations. Since we focus on the dynamics of rigid objects, our object-representation learning also uses the prior that all pixels of a rigid object have a consistent locomotion. Object mask representation allows our framework to leverage the well-studied object detection methods (e.g. Mask R-CNN [9]) in the computer vision area.
>
> Most generic object detection methods [10] and disentangled object-like representation learning approaches with priors built into some “general-purpose” objectives are supervised learning and require a large labeled training datasets, and do not perform action-conditioned relational reasoning for learning generalized dynamics. Few self-supersived or unsupervised approaches, like DRNET[11], \Beta-VAE [12] and DDPAE [13], are proposed for learning disentangled representations, but currently do not seem to work well in complex Atari-like domains. In short, we feel it is sensible to explore another direction for learning object-like representation and their dynamics from raw visual input.
>
> Q: When running your experiments, do you report results averaged over multiple runs? - Figure 4+C7: why does the x-axis start at 2000? - I don't think Figure 5 is really necessary - All figures: your captions could be improved by giving more information about what their figure presents. E.g. in Figure C7 I have no idea what the curves correspond to. Sure it's accuracy, but for which task? How many runs? Is it a running average? Etc. - Where are the test curves? Or are all curves test curves? - Your usage of \citep and \citet, (Author, year) vs Author (year), is often inconsistent with how the citation is used.
>
> A: To make the training process more efficient and stable in deep learning, there are usually a certain number of frames (2000 frames in our experiment) collected for populating the training buffer before learning start, which is also adopted in RL algorithms such as [14]. Thus, the first 2000 iterations are only used to collect an initial dataset and the learning process starts at iteration 2001. All the figures are test curves. Figure C7 plots the learning curves for the dynamics prediction in unseen Monster Kong environments . The curves with "Agent" notation illustrate the learning processes for the dynamics of the agent, while those with "All" notation indicate the learning curves of all dynamic objects.

---

> ### Author Response · Authors · 2018-11-28
> **Response to Reviewer 1 (connected to the previous response)**
>
> Q: The writing of this paper makes it a bit hard to understand what the novel contributions of this paper are, and how the proposed method should go beyond the two problems that it solves. In general, there are many phrasings that would benefit from being rewritten more concisely; it would help with clarity, since the proposed model has a multitude of different parts with sometimes long names.
>
> Experimentally, there are many parts to the proposed model, and while it is clear what each of them achieves, it is unclear how necessary each of the parts are, and how sensitive the model is to any part being (possibly slightly) incorrect.
>
> The proposed method is tested on, presumably, RL environments; yet, no RL experiments are performed, so there is no way of knowing if the proposed model is actually useful for planning (there are instances of model-based methods learning acceptable models that are just wrong enough to *not* be useful to actually do RL or e.g. MCTS planning).
>
> A: We have addressed these concerns in the general response to all reviewers.
>
> References:
> [1] BPL Lo and SA Velastin. Automatic congestion detection system for underground platforms. In Intelligent Multimedia, Video and Speech Processing, 2001. Proceedings of 2001 International Symposium on, pp. 158–161. IEEE, 2001.
> [2] Dar-Shyang Lee. Effective gaussian mixture learning for video background subtraction. IEEE Transactions on Pattern Analysis &amp; Machine Intelligence, (5):827–832, 2005.
> [3] Xiaowei Zhou, Can Yang, and Weichuan Yu. Moving object detection by detecting contiguous outliers in the low-rank representation. IEEE Transactions on Pattern Analysis and Machine 427 Intelligence, 35(3):597–610, 2013.
> [4] Xiaojie Guo, Xinggang Wang, Liang Yang, Xiaochun Cao, and Yi Ma. Robust foreground detection using smoothness and arbitrariness constraints. In European Conference on Computer Vision, pages 535–550. Springer, 2014.
> [5] Lucia Maddalena, Alfredo Petrosino, et al. A self-organizing approach to background subtraction for visual surveillance applications. IEEE Transactions on Image Processing, 17(7):1168,433 2008.
> [6] Watters, Nicholas, et al. "Visual interaction networks." arXiv preprint arXiv:1706.01433 (2017).
> [7] Wu, Jiajun, et al. "Learning to see physics via visual de-animation." Advances in Neural Information Processing Systems. 2017.
> [8] Vijayanarasimhan, Sudheendra, et al. "Sfm-net: Learning of structure and motion from video." arXiv preprint arXiv:1704.07804 (2017).
> [9] He, Kaiming, et al. "Mask r-cnn." Computer Vision (ICCV), 2017 IEEE International Conference on. IEEE, 2017.
> [10] Liu, Li, et al. "Deep learning for generic object detection: A survey." arXiv preprint arXiv:1809.02165 (2018).
> [11] Denton, Emily L. "Unsupervised learning of disentangled representations from video." Advances in Neural Information Processing Systems. 2017.
> [12] Higgins, Irina, et al. "beta-vae: Learning basic visual concepts with a constrained variational framework." International Conference on Learning Representations. 2017.
> [13] Hsieh, Jun-Ting, et al. "Learning to Decompose and Disentangle Representations for Video Prediction." arXiv preprint arXiv:1806.04166 (2018).
> [14] Mnih, Volodymyr, Kavukcuoglu, Koray, Silver, David S, and Rusu, Andrei A. et al. Human-level control through deep reinforcement learning. Nature, 2015

---

### Official Review · AnonReviewer2 · 2018-11-03
**Nice results but difficult to understand**

**Rating:** 4
**Confidence:** 4

**Review:**

This paper proposes a new architecture for learning dynamics models in 2D Atari-like game words. The architecture includes multiple layers of abstraction: a “motion detection” level, which looks at which pixels change over time in order to guess at which parts of the image are in the foreground or not; a “instance segmentation” level, which segments the foreground into regions and instances; and a “dynamics learning” level, which learns the dynamics of object instances using a interaction network-style approach.

Pros:
- Impressive-looking dynamics predictions in Atari-like games.
- An object-based prediction model, which could enable predictions about specific entities in the scene rather than holistic frame predictions.

Cons:
- Very complicated and difficult-to-understand architecture.
- No ablation studies to validate different components of the architecture.
- No validation in a model-based RL or control setting.
- Experiments are only done on one-step predictions, rather than long-term rollouts.

Quality
---------

The quality of the predictions seems quite high (based on Figure 6 and the results tables), though there are a number of opportunities to further strengthen the evaluation and analysis:

- I wish that there were more than a single figure of qualitative results to go on. I highly recommend that a revision include a link to a video showing more predictions over time for each environment, ideally with comparisons to the other baselines as well.
- The introduction of the paper motivates the learning of the model in terms of model-based RL, however, the model is not actually used in a model-based RL setting. It would be nice to see at least a simple validation that the model can be used with an off-the-shelf planner to solve one of the games which are evaluated in the paper. If it cannot, then that limits the significance of the model.
- As far as I can tell, all the results reported in the tables are based on one-step predictions only. While it is great to show that even in this regime the other models struggle, it would be even better if results could be reported for longer rollouts (i.e., taking the model outputs and feeding it back in as input, and repeating this procedure say 50 steps into the future). Models are not particularly useful in a MBRL setting if they can only be used to predict a single timestep, so it is important to validate that longer-term predictions can be made as well.

Overall the literature review is reasonably solid, but I am not sure the citations in the opening sentence are quite appropriate as model-based DRL has been around for longer than 2017 (see for example [1-3]). Moreover, Chiappa et al (2017) only learns a model and does not use it for planning, so I am not sure it is quite appropriate as a citation for MBRL.


Clarity
--------

Unfortunately, I had a very hard time understanding how exactly the architecture works and I felt like there were a lot of details missing. I am not confident that I would be able to reproduce the architecture from reading the paper alone. Below, I will list some of the specific points where I was confused, but I think overall the paper needs to be substantially reorganized in order to be clearer as to how the architecture actually works.

More broadly, I think some of my confusion stems from the fact that there are very similar computations occurring across the three levels of abstraction but the paper does not really make it clear how these computations relate to one another or how they are similar/different. For example, in the “dynamics learning” level there are modules for performing object detection and instance localization. But then in the “instance segmentation” level, there are similarly modules for detecting and masking out instances. It is not clear to me why this needs to be done twice?

In general, I would *strongly* recommend including at least in the appendix an algorithm box that sketches out the computational graph for the whole architecture (not in as much detail as the existing algorithm boxes, but in more detail than what is given in Figure 1).

Specific places where I was confused:

- Where do the region proposals (P) come from?
- If I’m understanding correctly, the variable M is used multiple times in multiple different ways. It seems to be produced from the “instance localization” module in the “dynamics learning” level, but also from the “dynamic instance segmentation network” in the “instance segmentation” level. Are these M different or the same?
- Where does F_foreground^(t) come from?


Originality
-------------

Overall, the idea of learning object-based transition models is not really new (and there are a few citations missing regarding prior work in this regard, e.g. [4-6]). However, there is yet to be an accepted solution for actually learning object-based models robustly and the present work seems to result in the cleanest separation between dynamic objects and background that I have seen so far, and is therefore quite original in that regard.

This paper appears to be quite similar to Zhu & Zhang (2018), with the main difference being additional functionality to handle multiple dynamic objects in a scene rather than just a single dynamic object. This is a fairly significant difference and the improvement over Zhu & Zhang (2018) seems quite large, so even though the papers seem quite similar on the surface I think the difference is actually quite substantial.

Significance
----------------

If it were clearer how to reproduce this paper, and if it could be shown to apply to a wider range of environments (e.g. the Atari suite, or even better the Sonic domains from the OpenAI Retro contest), then I believe this paper could be quite significant as it would open up new avenues for model-based learning in these domains. Unfortunately, however, it is not clear to me as the paper is currently written how well it would do on other 2D environments, thus limiting the significance. If the model only works on Monster Kong and Flappy Bird---neither of which are commonly used in the RL literature---then it has limited applicability to the rest of the model-based RL community. Similarly, as stated above, it is not clear how well the model will work with longer rollouts or in actual in MBRL settings, thus limiting its significance.

References
---------------

[1] Heess, Wayne, Silver, Lillicrap, Tassa, & Erez (2015). Learning Continuous Control Policies by Stochastic Value Gradients. NIPS 2015.
[2] Gu, Lillicrap, Sutskever, & Levine (2016). Continuous Deep Q-Learning with Model-based Acceleration. ICML 2016.
[3] Schmidhuber (2015). On Learning to Think: Algorithmic Information Theory for Novel Combinations of Reinforcement Learning Controllers and Recurrent Neural World Models. arXiv 2015.
[4] Wu, Yildirim, Lim, Freeman, & Tenenbaum (2015). Galileo: Perceiving Physical Object Properties by Integrating a Physics Engine with Deep Learning. NIPS 2015.
[5] Fragkiadaki, Agrawal, Levine, & Malik (2016). Learning visual predictive models of physics for playing billiards. ICLR 2016.
[6] Kansky, Silver, Mely, Eldawy, Lazaro-Gredilla, Lou, Dorfman, Sido, Phoenix, & George (2017). Schema Networks: Zero-shot Transfer with a Generative Causal Model of Intuitive Physics. ICML 2017.

---

> ### Author Response · Authors · 2018-11-28
> **Response to Reviewer 2**
>
> Thank you for your thoughtful review and suggestions.
>
> Q: - Very complicated and difficult-to-understand architecture.
> - No validation in a model-based RL or control setting.
> - Experiments are only done on one-step predictions, rather than long-term rollouts.
> - I highly recommend that a revision include a link to a video showing more predictions over time for each environment.
> - If it were clearer how to reproduce this paper, and if it could be shown to apply to a wider range of environments (e.g. the Atari suite, or even better the Sonic domains from the OpenAI Retro contest),
>
> A: We have addressed these concerns in the general response to all reviewers.
>
> Q: But then in the “instance segmentation” level, there are similar modules for detecting and masking out instances. It is not clear to me why this needs to be done twice?
>
> A: Although the network structure of this level is similar to Object Detector in the level of dynamics learning, we still add the “instance segmentation” level because it improves the stability of concurrently learning object representations and dynamics model. This level learns the coarse-grained object masks based on the spatial-temporal consistency of locomotion and appearance pattern, and the learned object masks are then used as proposal regions to persistently guide the object-oriented dynamics learning at the more fine-grained level, which can fine-tune the object representations, but not in a dramatic way.
>
> Q: Where do the region proposals (P) come from? - If I’m understanding correctly, the variable M is used multiple times in multiple different ways. It seems to be produced from the “instance localization” module in the “dynamics learning” level, but also from the “dynamic instance segmentation network” in the “instance segmentation” level. Are these M different or the same? - Where does F_foreground^(t) come from?
>
> A: P are the region proposals of dynamic instances for guiding the object representation learning at the “dynamics learning” level, which are learned by the more abstracted and coarse-grained learning level (i.e., “dynamic instance segmentation” level). Similarly, F_foreground^(t) are foreground region proposals for guiding the learning of the “dynamic instance segmentation” level, which are computed at the more abstract learning level, i.e., “motion perception” level. The variable M in the original paper is overloaded. The object masks M in the “instance segmentation” level become region proposals P at the “dynamics learning” level, guiding the learning of the object masks M at this level. We have corrected these notations to avoid confusion and rewrote the methodology to make it more clear. More details can be found in the revised version.
>
> Q: Overall, the idea of learning object-based transition models is not really new (and there are a few citations missing regarding prior work in this regard, e.g. [4-6]). However, there is yet to be an accepted solution for actually learning object-based models robustly and the present work seems to result in the cleanest separation between dynamic objects and background that I have seen so far, and is therefore quite original in that regard.
>
> A: We have properly cited the papers according to the review’s suggestion. [5,6] assumed that the object localization and tracking, or the object representations are given, and directly used them to learn the object-based dynamics model. [4] proposed to use a realistic physics engine called Bullet physics engine to perceive physical object properties. Unlike them, our novelty lies in developing a self-supervised neural network framework that automatically learns object representations and object-based dynamics from raw visual observations and demonstrating the generalization ability of this framework over novel environments with multiple dynamic objects and different object layouts.
>
> [4] Wu, Yildirim, Lim, Freeman, & Tenenbaum (2015). Galileo: Perceiving Physical Object Properties by Integrating a Physics Engine with Deep Learning. NIPS 2015.
> [5] Fragkiadaki, Agrawal, Levine, & Malik (2016). Learning visual predictive models of physics for playing billiards. ICLR 2016.
> [6] Kansky, Silver, Mely, Eldawy, Lazaro-Gredilla, Lou, Dorfman, Sido, Phoenix, & George (2017). Schema Networks: Zero-shot Transfer with a Generative Causal Model of Intuitive Physics. ICML 2017.

---

> > ### Comment · AnonReviewer2 · 2018-12-03
> > **Response to authors**
> >
> > Thank you for your response and for including the additional experiments with longer rollouts, evaluation on an additional environment, the video demonstrating the approach, and further details in the paper. I believe this does make the paper quite a bit stronger. However, I unfortunately feel two of my major points have not been fully addressed (the first one below being more important than the second) and therefore I am not inclined to change my score.
> >
> > First, while the paper is indeed clearer than before, I still do not feel like the architecture has been explained clearly enough for it to be accepted. For example, while I appreciate Algorithm 1, it does not include any information about how the Instance Segmentation level feeds into the Dynamics Learning level. Instead, the answer is buried at the end of the "Prediction and Training Loss" section on page 6: the Dynamics Learning level includes an additional loss term that computes the L2 loss between the masks proposed at the Dynamics Learning level and the Instance Segmentation level. As I mentioned in my original review, I strongly recommend including pseudocode or an algorithm box for the *entire* architecture (not just Dynamics Learning level), as it is very difficult to otherwise understand how all the parts interact.
> >
> > Second, while I am sympathetic to the fact that model learning in and of itself is difficult, if the point of the model is to be used within an RL system then it really should be be validated against an RL system. Small model errors that might seem insignificant when judged via L2 loss (or whatever metric is chosen) may actually be very problematic when trying to use the model in the context of a larger system. This issue was raised both by myself and by R1, and I do not feel like the response that "applications of the learned dynamics model are not the focus of this paper but remain to be the future work" really addresses this concern. While I don't think this issue is absolutely necessary for acceptance (certainly other model learning papers have not always included an evaluation in a model-based RL system), I think having this would offset some of the concerns about the system being overly complex or specific and would make the paper significantly stronger.

---

> > > ### Author Response · Authors · 2018-12-11
> > > **Response to Reviewer 2**
> > >
> > > Thank you for your constructive suggestions. The entire architecture of our multi-level abstraction framework can be summarized as follows:
> > >
> > > Step 1: Initialization. Initialize the parameters of all neural networks with random weights respectively.
> > >
> > > Step 2: Motion Detection Level. Perform foreground detection to produce dynamic region proposals, which potentially have moving objects
> > >
> > > Step 3: Instance Segmentation Level. Train the dynamic instance segmentation network (including Instance Splitter and Merging Net) by minimizing L_DIS, which includes a proposal loss to focus the dynamic instance segmentation on the dynamic region proposals from Step 2.
> > >
> > > Step 4: Dynamic learning Level. Train the dynamics learning network (whose forward process is shown as Algorithm 1) by minimizing L_DL, which includes a proposal loss to utilize the dynamic instance proposals generated by the trained dynamic instance segmentation network in Step 3 to facilitate the learning of Object Detector.
> > >
> > > We will add these descriptions in the next version of our paper.

---

### Author Response · Authors · 2018-11-28
**General response to all reviewers**

We thank all reviewers for their feedback and thoughtful comments and suggestions, which are helpful for improving the quality of our paper. In this updated paper, we have revised our manuscript according to their comments and suggestions. Below, we describe in detail how we have modified our paper to address the reviewers’ feedback.

1. We refined the presentation of our method, and added additional descriptions to illustrate the high-level intuition of the architecture design and clarified our main contributions.

2. We compared our model with baselines in terms of the long-term predictions in unseen environments.

3. We add a video for better perceptual understanding of the prediction performance in unseen environments.

In addition, we take this opportunity to emphasize the main contributions of this paper:

1. We propose a novel self-supervised, object-oriented dynamics learning framework to enable sample-efficient learning and zero-shot generalization over novel environments that have multiple controllable and uncontrollable dynamic objects and different static object layouts.

2. Our approach takes a step towards interpretable deep learning and disentangled representation learning. It learns disentangled representations and visually and semantically interpretable knowledge, which contributes to understanding the logic behind the dynamics prediction and opens the avenue for further researches on object-based planning, object-oriented model-based RL, and hierarchical learning.

3. We provide a general multi-level framework for learning object-based dynamics model from raw visual observations, which offers opportunities to easily leverage the well-studied object detection methods (e.g. Mask R-CNN [He et al., 2017]) in the computer vision area.

Our main objective lies in learning generalizable and interpretable dynamics from raw visual observations, which is a general-purpose task for AI and potentially benefit a broad range of domains. For example, the learned dynamics model can guide the exploration of model-free RL [Chiappa et al., 2017], be used with existing policy search or planning methods (e.g., MCTS and MPC) [Finn et al., 2017], or directly plugged into an end-to-end policy network integrating model-free and model-based path [Weber et al., 2018]. The prediction error of our dynamics model can be used as signals for curiosity-driven exploration [Pathak et al., 2017]. Our learned object representations can be leveraged to design effective heuristic reward functions (like the distance-based rewards [Srinivas et at., 2018]) to facilitate model-free RL, or used to set subgoals in hierarchical RL [Kulkarni et al., 2016]. However, these applications of the learned dynamics model are not the focus of this paper but remain to be the future work.

We also want to address the general concern about universality of our approach. The assumption of our method is that the environment only contains rigid objects and has no camera motion. Under this assumption, we choose another game Freeway from Atari Game to test our model and get similar performance with Monsterkong and Flappy Bird. To test generalization ability, we use first 1800 frames for training and the last 200 frames for testing. For the training frames, our model achieves 0.80, 0.91, and 0.94 for 0-error, 1-error, and 2-error accuracy, respectively. For the testing frames, our model achieves 0.79, 0.89, and 0.94 for 0-error, 1-error, and 2-error accuracy, respectively.

[1] He, Kaiming, et al. "Mask r-cnn." Computer Vision (ICCV), 2017 IEEE International Conference on. IEEE, 2017.
[2] Chiappa, Silvia, et al. "Recurrent environment simulators." arXiv preprint arXiv:1704.02254 (2017).
[3] Finn, Chelsea, and Sergey Levine. "Deep visual foresight for planning robot motion." Robotics and Automation (ICRA), 2017 IEEE International Conference on. IEEE, 2017.
[4] Racanière, Sébastien, et al. "Imagination-augmented agents for deep reinforcement learning." Advances in Neural Information Processing Systems. 2017.
[5] Pathak, Deepak, et al. "Curiosity-driven exploration by self-supervised prediction." International Conference on Machine Learning (ICML). Vol. 2017. 2017.
[6] Srinivas, Aravind, et al. "Universal Planning Networks." arXiv preprint arXiv:1804.00645 (2018).
[7] Kulkarni, Tejas D., et al. "Hierarchical deep reinforcement learning: Integrating temporal abstraction and intrinsic motivation." Advances in neural information processing systems. 2016.

---

### Author Response · Authors · 2018-12-01
**Additional modular test to better address the reviews' concerns**

We conduct modular test to better understand the contribution of each abstraction level (the detailed results are shown in https://github.com/maop2018/maop-video/blob/master/MAOP.pdf ).  First, we investigate whether the level of dynamics learning can learn the accurate  dynamics model when the coarse region proposals of dynamic instances are given. We remove the other two levels and replace them by the artificially synthesized coarse proposals of dynamic instances to test the independent performance of the dynamics learning level. Specifically, the synthesized data are generated by adding standard Gaussian or Poisson noise on ground-true dynamic instance masks (Figure 1). As shown in Table 1, the level of dynamics learning can learn accurate dynamics of all dynamic objects given coarse proposals of dynamic instances. Similarly, we also test the independent performance of the dynamics instance segmentation level. We replace the foreground proposal generated by the motion detection level with the artificially synthesized noisy foreground proposal. Figure 2 shows cases to demonstrate our learned dynamic instances in the level of dynamic instance segmentation, which demonstrates the competence of the dynamic instance segmentation level. Taken together, the modular test shows that each level of MAOP can independently perform well and has a good robustness to the proposals generated by the more abstracted level.

We also provide the detailed results on Freeway from Atari games, which has a large number of dynamic objects. To test generalization ability, we use first 1800 frames for training and the last 200 frames for testing. As shown in Table 2, our model outperforms the existing modeling methods in this domain. Note that only the ground-true location of the agent is accessible in Arcade Learning Environment, so we just show the quantitative prediction performance of the agent's dynamics. Actually, we observe that the predictions of other dynamic objects are also accurate by comparing the predicted with the ground-true images, as shown in Figure 3. The validation results on Freeway demonstrate that our model is effective for the concurrent dynamics prediction of a large number of objects.

We will add these results in the next version of our paper. We would like to thank again for the reviews' suggestions.

---

### Meta-Review · Area_Chair1 · 2018-12-17
**an interesting formulation for 2D dynamics learning not clearly described**

**Confidence:** 5
**Recommendation:** Reject

**Metareview:**

This paper tackles a very valuable problem of learning object detection and object dynamics from video sequences, and builds upon the method of Zhu et al. 2018. The reviewers point out that there is a lot of engineering steps in the object proposal stage, which takes into account background subtraction to propose objects. In its current form, the writing of the paper is not clear enough on the object instantiation part, which is also the novel part over Zhu et al., potentially due to the complexity of using motion to guide object proposals. A limitation of the proposed formulation is that it works for moving cameras but only in 2d environments. Experiments on 3D environments would make this paper a much stronger submission.